**communications** engineering

# Deep learning reduces sensor requirements for gust rejection on a small uncrewed aerial vehicle morphing wing
Kevin P. T. Haughn [1] ✉, Christina Harvey [2] & Daniel J. Inman [3]

Uncrewed aerial vehicles are integral to a smart city framework, but the dynamic environments above and within urban settings are dangerous for autonomous flight. Wind gusts caused by the uneven landscape jeopardize safe and effective aircraft operation. Birds rapidly reject gusts by changing their wing shape, but current gust alleviation methods for aircraft still use discrete control surfaces. Additionally, modern gust alleviation controllers challenge small uncrewed aerial vehicle power constraints by relying on extensive sensing networks and computationally expensive modeling. Here we show end-to-end deep reinforcement learning forgoing state inference to efficiently alleviate gusts on a smart material camber-morphing wing. In a series of wind tunnel gust experiments at the University of Michigan, trained controllers reduced gust impact by 84% from on-board pressure signals. Notably, gust alleviation using signals from only three pressure taps was statistically indistinguishable from using six pressure tap signals. By efficiently rejecting environmental perturbations, reduced-sensor fly-by-feel controllers open the door to small uncrewed aerial vehicle missions in cities.

Although both the public sector and defense agencies are interested in urban uncrewed aerial vehicle (UAV) mission performance, fixed winged aircraft are still incapable of adapting to the complex aerodynamics within a city environment[1–3]. Currently, the most dynamic environments are dominated by multirotor flight vehicles; however, the highly maneuverable and responsive quadrotor design suffers from substantial weight and power constraints, limiting the operational range and on-board computational capabilities needed for autonomy[4–7]. Current fixed wing UAVs have greater range but are not as maneuverable[8]. Counter to both rotorcraft and traditional fixed wing UAV design, birds can adapt their wing shape as the environment changes to achieve both efficient and maneuverable flight[9,10]. This ability supports birds of prey in navigating through complex environments[11], or rejecting perturbations in a gusty environment[12,13]. UAVs can achieve a similar adaptive gust rejection by changing the shape of their wings with camber morphing (Fig. 1a).

Wing morphing brings several challenges regarding mechanical complexity and compliance with the weight and volume constraints of small UAV design. Recent advances in smart materials offer a clever way to address these challenges[14,15]. Macro-fiber composites (MFC) have been used for bio-inspired soft robotics and can act as both the skin and actuator of a camber-morphing wing[16,17]. By rapidly changing the wing's curvature,

MFCs can actively reduce the aerodynamic forces experienced during gusts without the mechanical complexity associated with large scale shape changes. In addition, the smooth shape change offered by MFC camber-morphing improves aerodynamic efficiency, speed, weight reduction, and overall control authority when compared to traditional rigid flap actuation methods[18–20]. However, MFCs suffer from hysteresis, creep, and inconsistent performance under out-of-plane loading. These challenges informed our autonomous gust alleviation (GA) controller design for a camber morphing wing with three active MFC sections (Fig. 1b).

Autonomous gust rejection is a key part of the puzzle that must be achieved to enable small, fixed wing UAVs to complete missions in complex aerodynamic environments, thus expanding the operational range compared to their quadrotor counterparts. Perturbations, such as gusts, impact flight performance and complicate tracking of predefined trajectories[21]. This is especially true for small UAVs due to their lightweight nature. Historically, gust response requires a pilot or autopilot to respond to a perturbation with an antagonistic action[22,23]. However, these corrections occur after the external force has already perturbed the aircraft, and pilot reaction times typically fall between 0.4 s and 1.3 s after a perturbation signal before providing an input reaction[24]. This may compromise mission success when strict altitude caps are in place, such

[1]U.S. Army Research Laboratory; Aberdeen Proving Ground, Aberdeen Proving Ground, MD, USA. [2]Department of Mechanical and Aerospace Engineering, University of California Davis, Davis, CA, USA. [3]Department of Aerospace Engineering, University of Michigan, Ann Arbor, MI, USA. ✉e-mail: kevin.p.haughn.civ@army.mil

**Fig. 1 | Natural flyers use wing shape morphing to reject gusts. a** Inspired by how birds change the shape of their wings to adjust for environmental changes, we implemented a trailing edge camber morphing mechanism. **b** The morphing wing consisted of 3 active sections driven by macro-fiber composites (MFC). A rigid wing acting as a gust generator was mounted at quarter chord 30 cm upstream of the morphing wing with three active camber morphing sections within the University of Michigan 30 cm × 30 cm wind tunnel. **c** The morphing wing was designed with six pressure taps to sense gusts. **d** The gust generator deflected upwards (yellow) and downwards (green) at varying degrees (depicted by opacity) to create a variety of velocity wakes, **e** the magnitude of which was quantified with particle image velocimetry.

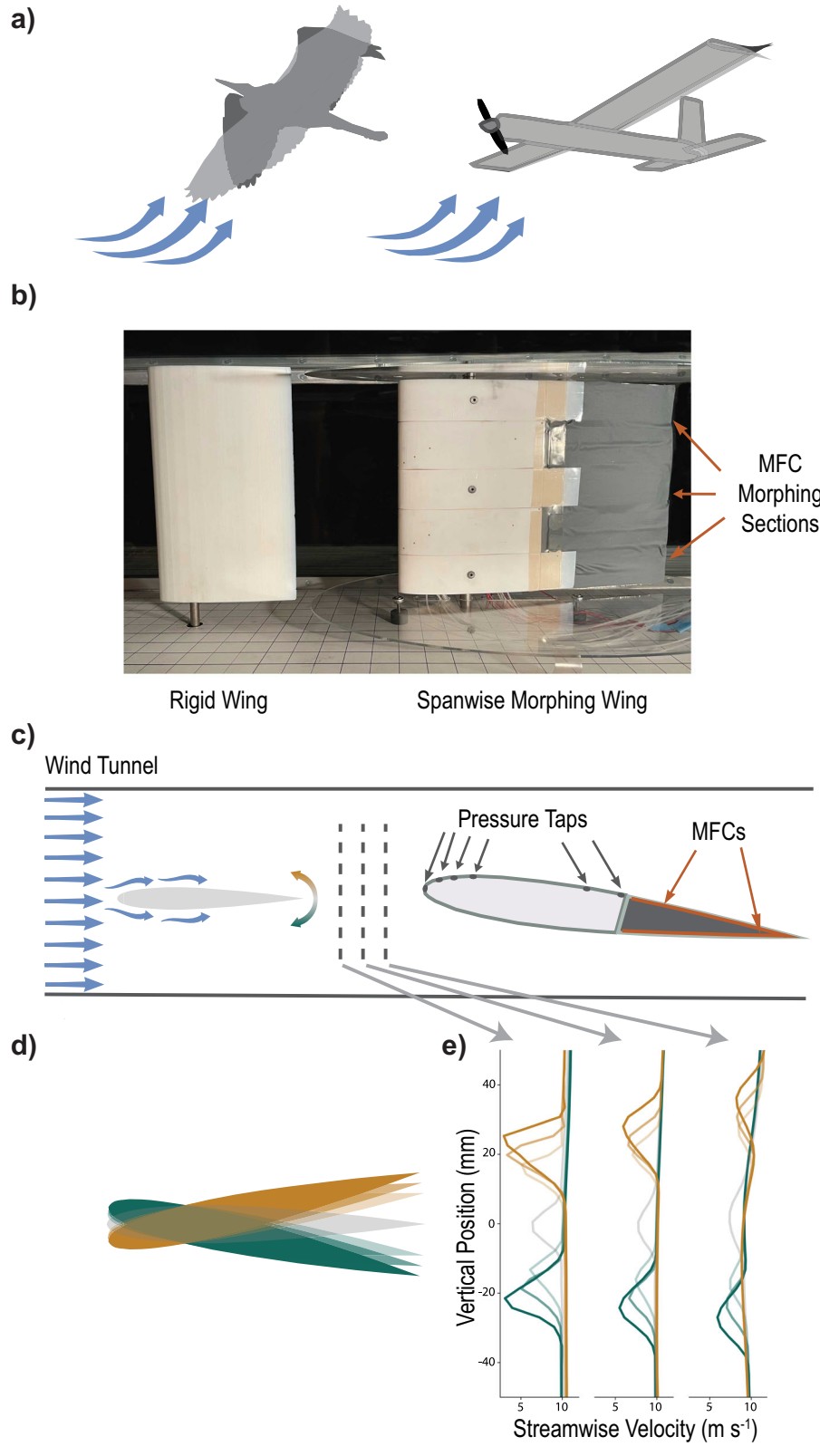

as during nap-of-the-earth flight[25]. Autopilot systems following classical control theory have used traditional control surfaces with strain gauges for feedback to achieve 50% gust load and flight ride quality improvement[23]. Recently this response has been improved to 80% when assuming a Doppler light detection and ranging (LIDAR) system was available to provide a preview of incoming gusts[26]. Instead of responding to a perturbation after it occurs, or spending computation and weight resources on LIDAR systems to look ahead for future perturbations, our fly-by-feel (FBF) active GA senses environmental changes on the wing in real time, beginning the initial morphing reaction in as little as one discrete timestep (0.05 s) to mitigate unintended changes in aerodynamic forces during a gust.

Successful adaptation, such as that provided by GA, relies on an accurate representation of the changing environment[27–29]. FBF is a biologically inspired

paradigm that uses distributed sensors to inform UAVs of environmental changes[29–34]. Recently, FBF achieved up to 76% mean gust rejection on a servo-driven camber morphing wing by using incremental nonlinear dynamic inversion with quadratic programming and virtual shape functions (INDI-QP-V), incorporating sixteen on-board piezoelectric pressure sensors to detect changes in the airflow for state inference as well as fourteen fiberoptic cables, twelve strain gauges, and a wing root mounted camera to detect camber deflection with proprioceptive modeling[35]. However, the expansive sensing networks used to inform decision making through proprioception and state inference add weight and challenge the computational power capabilities offered by small UAVs[5–7,28,36]. Instead of relying on vast amounts of sensory data for decision making, we used intelligent controller design to determine if fewer sensors could be used to achieve GA while reducing computational cost. The model-based controllers often used for GA require highly accurate predictions to achieve sufficient control because any errors produced prior to action selection propagate through the controller. This dramatically increases computational costs[27,37–39]. Alternatively, model-free deep reinforcement learning (DRL) can train neural networks to make action decisions directly from raw sensor inputs without using dynamics or state inference models[40,41]. Proximal policy optimization (PPO) is a DRL algorithm that has shown to account for MFC hysteresis and produce effective camber control in a morphing airfoil[42,43]. For this reason, we used PPO to develop the GA policies (i.e., controllers) directly from three different sensor combinations (Supplementary Fig. 1). Controllers were trained to make decisions in a gusting wind tunnel environment based on pressure signals provided by up to six pressure taps installed on the top surface of the morphing wing (Fig. 1c).

Most successful DRL applications are trained in simulation due to the repetitive nature of DRL's trial-and-error training format[44,45]. However, accurately simulating complex, gusty environments requires large computational time and cost[46,47]. We avoided the computational costs as well as the uncertainty associated with simplified approximation by training directly on the physical hardware environment. Although training in the physical hardware space offers unique challenges, we found success using methods emphasizing efficiency and autonomy in state-action exploration through a pseudo-episodic training method[48,49]. This training format requires an automatic transition between episodes. Therefore, we adapted methods previously established in the literature to automate a gusting environment[29,35,36]. By deflecting a rigid wing, mounted in a wind tunnel upstream of our morphing wing, we exposed the morphing wing to a broad range of repeatable gusts during training to facilitate thorough exploration of the dynamic environment's state and action spaces (Fig. 1d). Exploration is crucial for developing a robust controller capable of effectively rejecting the various degrees of perturbation experienced in a city. Therefore, during training the gust generator induced a variety of wakes representative of the updrafts and downdrafts experienced when flying over the complex street systems between buildings (Fig. 1e). Autonomously rejecting these types of gusts with reduced-sensor FBF will open the door to urban flight for fixed wing UAVs.

## Results
### Gust impact and reduction
The gust generator used in this wind tunnel environment perturbed the local angle of attack for the incoming airflow in a manner analogous to common flight situations in natural and urban environments (Supplementary Fig. 2). The controller experienced the gusts as instantaneous changes in wind speed and direction, similar to a sharp-edged gust model (see materials and methods). This model is often used to imitate an aircraft encountering an updraft, as found between two buildings, resulting in a change in lift[21,23,50,51]. The magnitude of gust-generated lift that was rejected by the active morphing wing was termed the gust rejection percentage (GRP) defined as:

$$\text{GRP}(t) = \left(1 - \frac{|\Delta L_C(t)|}{|\frac{1}{T}\sum_{t=0}^{T} \Delta L_B(t)|}\right) \times 100\%. \tag{1}$$

GRP was measured as a percentage difference between the change in lift during active morphing control, $\Delta L_C$, and the baseline average change in lift, $\Delta L_B$, produced by the wing when unactuated over the duration of the gust, $T$ (Fig. 2a, b). To replicate common scenarios experienced during city flight, tests were conducted at three different flight conditions (low-lift, medium-lift, and high-lift) for three gust magnitudes (mild, moderate, and strong) in two directions (upward and downward) (Supplementary Table 1). Although the high-lift condition experienced smaller gust impact (5% change in lift), the medium-lift and low-lift conditions experienced much larger ranges and magnitudes of gust impacts (28% and 29% change in lift, respectively). To define the stability and robustness of the trained neural network policies, we trained a total of twenty (20) policies and repeated gust alleviation performance tests ten (10) times for each gust condition (6), resulting in 1200 gust rejection wind tunnel tests. We quantified a controller's consistency between individual test iterations, gust conditions, and trained policies using the average standard deviation (STD) of the settled GRP between tests while holding all other factors constant. The settled GRP was consistent between test iterations for a single policy at each gust condition (high-lift: STD = 4.9%; medium-lift: STD = 2.3%; low-lift: STD = 2.5%) (Fig. 2c), but the average settled GRP performance of individual trained policies was less consistent between gust conditions (high-lift: STD = 10.5%; medium-lift: STD = 21.4%; low-lift: STD = 19.0%) (Fig. 2d). However, the average settled GRP was consistent between trained policies for each gust condition (high-lift: STD = 8.2%; medium-lift: STD = 7.5%; low-lift: STD = 5.7%) (Fig. 2e).

We repeated the training and testing process described above to measure GRP for three sensor configurations: one, three, and six chordwise distributed pressure taps (Fig. 3a). This resulted in 3600 gust rejection wind tunnel tests in total. We found the number of pressure taps used for state observation significantly affected the trained GA controller performance.

### Diminishing effect of rearward sensors
We used the settled GRP from each test to calculate the mean gust rejection percentage for each pressure tap configuration and gust condition (Fig. 3b–d). Controllers using all six pressure taps consistently achieved large mean gust rejections for each flight condition (high-lift: 84%; medium-lift: 84%; low-lift: 86%) relative to the respective gust-generated change in lift. When we reduced the number of signals informing the DRL algorithm to only use one pressure tap, we found a significant reduction in the gust rejection performance (high-lift: $P = 0.006$; medium-lift: $P < 0.001$; low-lift: $P < 0.001$). However, when using only three pressure taps, we found an insignificant effect on the gust rejection compared to the six-tap case for all tested flight conditions (high-lift: $P = 0.40$; medium-lift: $P = 0.32$; low-lift: $P = 0.67$). This result indicates that the increased complexity of the six-tap input did not yield additional improvements in gust rejection performance beyond the three-tap construction. In fact, for the medium-lift flight condition, the three-tap configuration achieved greater, although not significantly greater, mean gust rejection.

The mean GRP is only part of the puzzle. Performance consistency is important if this approach is to provide safe and reliable flight control for future UAVs. Therefore, we directly considered the uncertainty of our gust rejection metric using the standard deviation of the settled GRP distributions (Supplementary Fig. 3a–c). We found that the one-tap configuration was significantly less consistent than the controllers using more pressure taps (high-lift: $P = 0.001$; medium-lift: $P < 0.001$; low-lift: $P < 0.001$). Like the mean results, we found no significant difference between the consistency of the six-tap and three-tap configurations (high-lift: $P = 0.20$; medium-lift: $P = 0.91$; low-lift: $P = 0.46$). Note that the standard deviations were small relative to the gust-generated change in lift (one tap: 15%, three taps: 14%, six taps: 12%), suggesting that the active morphing gust rejection was overall quite consistent for our implementation.

Timing is also a crucial component of perturbation response since a slower reaction would negate much of the benefit offered by the correction. The instantaneous change in lift produced by the sharp-edged gusting environment neglected the buildup in gust intensity typically found in

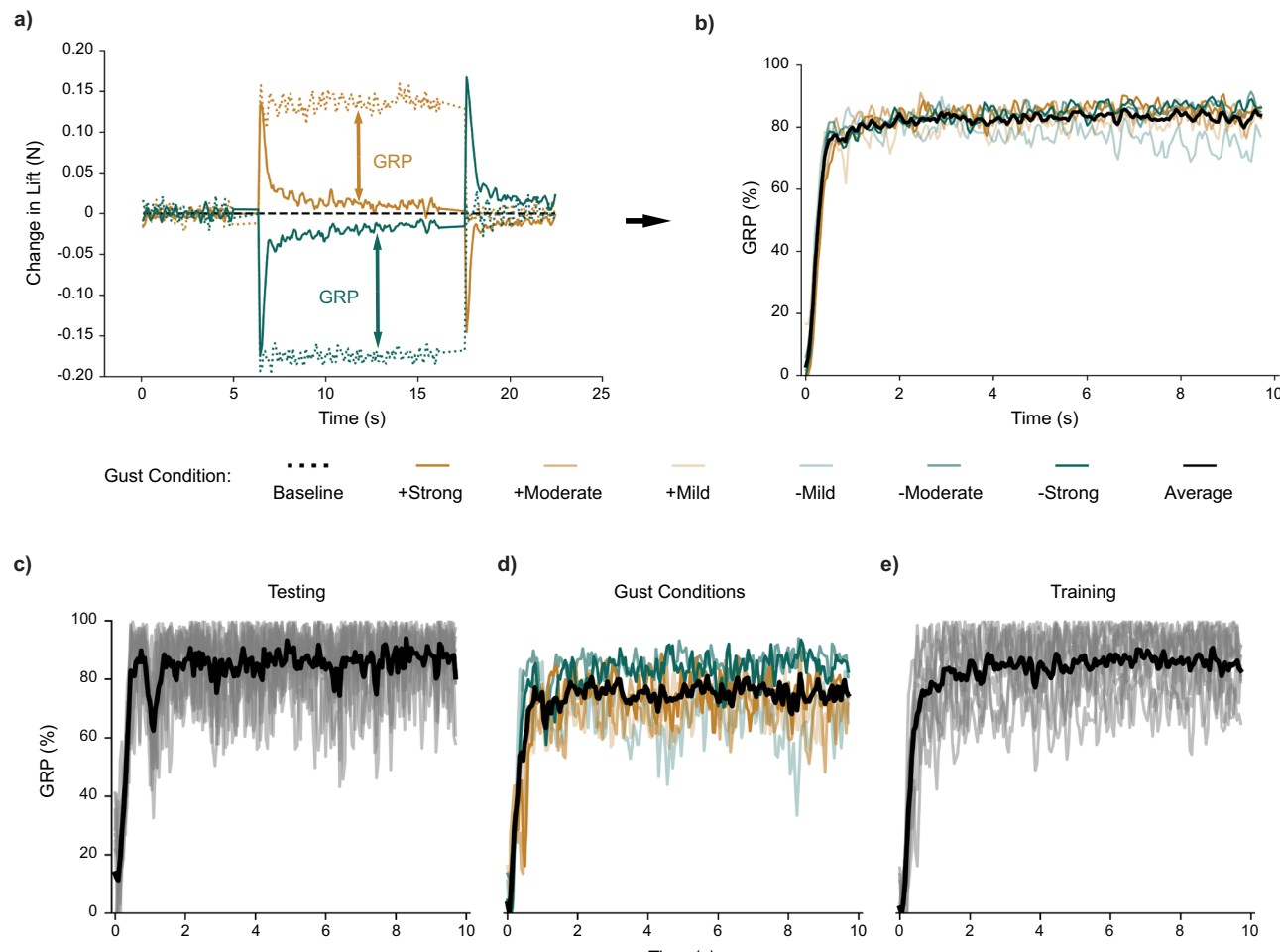

**Fig. 2 | Gust Rejection Percentage (GRP) provides a metric for controller performance and consistency.** With Proximal Policy Optimization, we trained 10 controllers using six pressure taps for gust alleviation in the high-lift flight condition gusting environment. **a** We quantified controller performance by comparing the change in lift of the actively controlled wing (solid line), $\Delta L_C$, with that of the inactive baseline (dotted line), $\Delta L_B$, where the magnitude of the arrows indicates GRP. **b** On average (n = 600), the learned controllers rejected more than 84% of the $\Delta L_B$ produced by the tested gusts. In addition, we measured consistency between tests, gust conditions, and trained controllers using the standard deviation between **c** ten (10) tests for one trained controller at one gust condition, **d** average gust responses for a single controller at each gust condition (6), **e** and the average responses at a single gust condition for each trained controller (10). Gust deflections included both upward (yellow) and downward (green) directions at three strengths (increasing with opacity).

nature, creating a challenging environment for controller response. Still, using rise time, we quantified the controllers' speed to comment on how reducing sensor count impacted the active responsiveness of the system (Fig. 3e). We found that the controller speed was not significantly affected by the pressure tap configurations ($P > 0.05$) for all flight conditions and was consistent with rise times established in previous work where DRL controllers showed to be faster than traditional feedback control methods for an MFC morphing wing[42] (Fig. 3f–h). However, the higher intensity gusts resulted in greater rise times, which suggests the limited discrete action space likely restricted controller speeds. Rise time uncertainty was considered using standard deviation, as done previously with gust rejection (Supplementary Fig. 3d–f).

Next, we explored the functional differences between the number of taps used and found that sensitivity of the pressure taps decreased towards the trailing edge of the wing (Fig. 4a), explaining the insignificant difference in performance between using three sensors and six sensors. The leading-edge pressure taps showed the greatest sensitivity for both positive and negative gust deflections, which is consistent with expectations as this region is usually responsible for the largest suction peak on lift producing airfoils. Comparing upward and downward gusts in the high-lift flight condition, the second pressure tap showed less sensitivity (27% reduction) during the downward gust than during the upward gusts. The third tap, however,

showed a steep reduction in sensitivity (83%) when experiencing a downward gust as opposed to an upward gust. Similar effects occurred in the other flight conditions as well (Supplementary Fig. 4).

**Downward gusts challenge sensing**

Despite the overall success, we found situations in which the controllers underperformed relative to the other tested gust conditions, including the mild downward gust during high-lift flight (Fig. 3b). For this condition, the wing morphing controller overcompensated by actuating the trailing edge to a magnitude appropriate for a larger change in lift (Fig. 4b). However, this effect did not occur for the mild upwards gust in the same flight condition. These results suggested that the controllers were less effective at differentiating between the magnitudes of downward gusts in this flight condition.

To investigate further, we used particle image velocimetry (PIV) to quantify the change in local flow velocity across the top surface of the morphing wing at each tested gust condition compared to the baseline neutral gust condition during high-lift flight (Fig. 4c). The mild upward gust condition (7.5° gust generator deflection) increased the flow velocity over the first three pressure taps. The mild downward gust (−7.5° gust generator deflection) reduced velocity at the leading edge of the wing. However, the change in velocity shifted from negative to positive near the third pressure

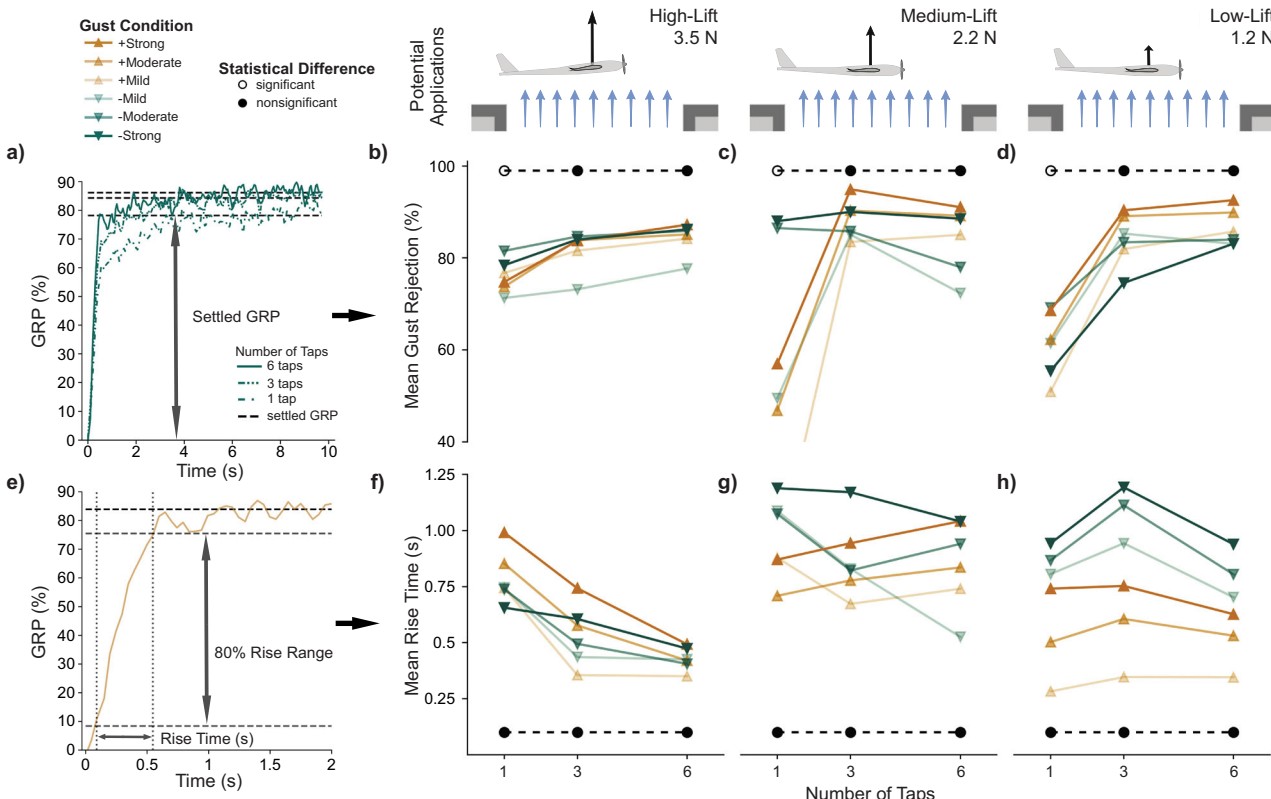

**Fig. 3 | The number of pressure taps significantly affected gust rejection performance. a** We used the settled gust rejection percentage (GRP) to measure controller effectiveness for each pressure tap configuration (1 tap: dashed-dot line; 3 taps: dash-dot-dot-dot line; 6 taps: solid line), at each gust direction (upward: yellow triangle; downward: green inverted triangle) and gust strength (increasing with opacity) for lift conditions (high-lift, medium-lift, and low-lift). **b–d** Controllers relying on a single pressure tap rejected a significantly smaller portion of the gust

than controllers using all six taps for all flight conditions (high-lift: $P = 0.006$, medium-lift: $P < 0.001$, low-lift: $P < 0.001$) as represented by open circles. However, the difference between using three pressure taps and six pressure taps was not significant for each flight condition (high-lift: $P = 0.40$, medium-lift: $P = 0.32$, low-lift: $P = 0.67$). **e** We quantified controller speed using rise time, the time needed to increase GRP from 10% to 90% of the settled GRP. **f–h** Varying the number of pressure taps did not significantly affect the rise time.

tap, producing a minimal pressure change. For the strong downward gust ($-12.5°$ gust generator deflection) there was a larger reduction of velocity at the leading edge of the wing, but the velocity change near the third pressure tap was still weak. Despite this, the trained controllers still achieved mean GRP values of above 73% for the three-tap and six-tap configurations in this challenging gust condition.

The strong downward gust during low-lift flight also produced disproportionately low performance relative to the other gusts within the same flight condition. In this case, the controller undershot the target, again suggesting it was difficult to distinguish between downward gust magnitudes. Interestingly, this gust was generated by a similar deflection angle ($-8°$) to that of the other challenging gust condition. This may provide insight into a challenging characteristic specific to our gust generating mechanism as opposed to a deficiency in the gust rejection controller design. The wake behind a deflecting wing produced changes in lift similar to those experienced during a vertical gust but generated additional streamwise aerodynamic effects (Fig. 1e) that are absent in traditional gust models.

## Discussion

Here we showed a FBF controller that does not require many sensors to effectively reject gusts. The learned controllers consistently achieved greater than 80% gust rejection without the computational and mechanical complexities associated with expansive distributed sensing networks. This suggests that the success of FBF aircraft need not depend on our ability to implement highly complex large scale distributed networks if we can effectively identify a reduced set of sensors that provides comparable performance. These results run counter to the big data mentality that is pervasive in deep learning and has recently driven sensor network design in

machine learning based distributed sensing applications, including FBF[30,31,52]. Like intelligent feature selection in deep learning, intelligent controller and sensor design can achieve reduced-sensor FBF, providing an efficient alternative to large-scale distributed sensing networks[53]. This reduces mechanical complexity and cost during fabrication as well as weight and computational requirements during operation. In addition, where human pilots naturally have a delayed initial reaction (0.4 s to 1.3 s) to gusts, FBF can begin changing shape in as little as a single timestep (0.05 s), and we showed that the controller speed was not impacted by reduced sensor input[24]. Further, we expect that optimizing the controller action space would provide a more rapid response. Our findings suggest that these cost-effective solutions can expand the mission scope of small, fixed-wing UAVs to increasingly dynamic environments. This creates the opportunity for numerous critical applications[8,54].

Incorporating reduced-sensor FBF UAVs for surveillance and disaster response will drastically improve safety for those living in large cities[4]. The range offered by fixed wing designs will provide greater coverage than that achieved by quadrotor designs, allowing them to provide broader surveillance or survey fire and earthquake scenes across the city for extended periods of time. This technology may prove particularly useful to first responders impeded by street traffic, communicating crucial information to improve efficiency and safety. Similarly, we can apply these methods to long range urban reconnaissance for soldiers encountering potentially dangerous situations.

Finally, the success of this model-free method promotes future intelligent aircraft designs for other complex maneuvers and environments where accurate models are not readily available. For example, similar hardware-based learning may produce controllers for morphing UAVs with

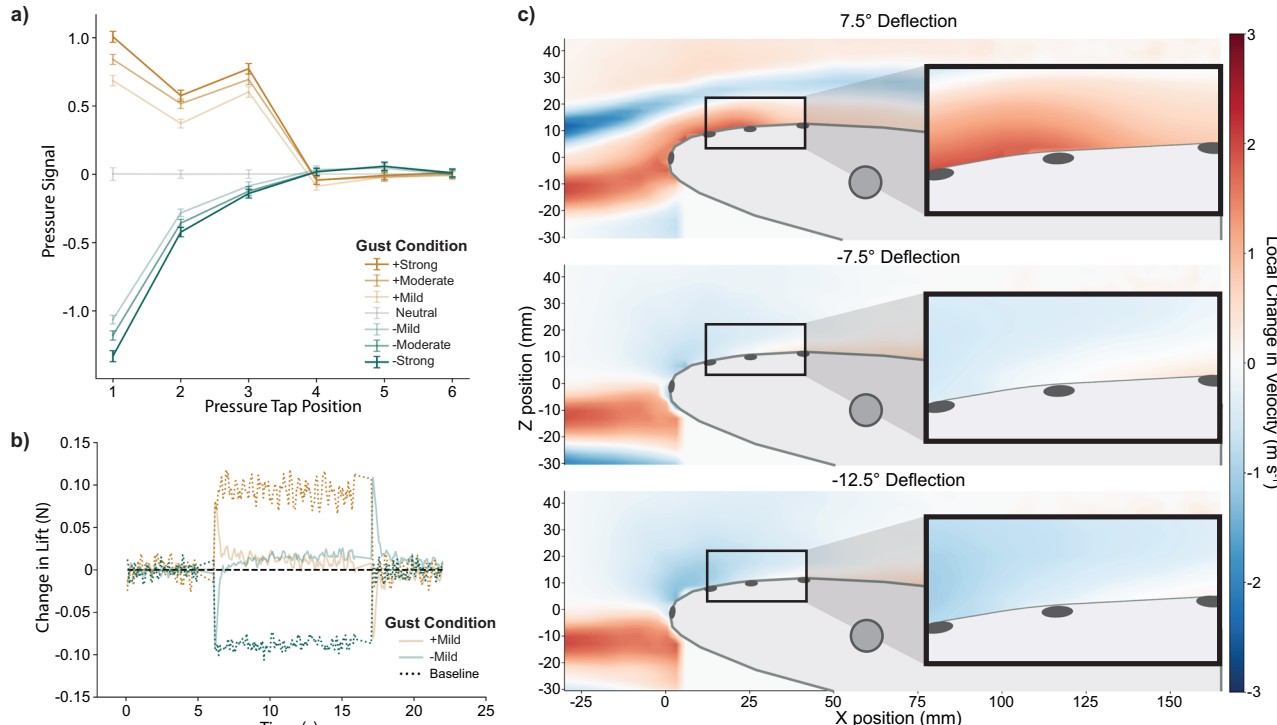

**Fig. 4 | The third pressure tap lost sensitivity during downward gusts for the high-lift flight condition. a** Although the first three pressure taps produced sensitive pressure signals (error bars represent 95% confidence intervals) for the upward (yellow line) gust deflections, the third pressure tap was much less sensitive to downward (green line) gusts (16.7%). **b** At the mild downward gust condition, the trained gust alleviation controllers using six pressure taps overshoot zero lift error. **c** Particle image velocimetry (PIV) showed the environmental change in the local

velocity experienced by the wing during different gusts. This change was measured by directly comparing the velocity at each position during a gust to that experienced during the neutral airflow. Blue represents a decrease in velocity at the specific position due to the gust generator, and red shows an increase in velocity. The change in local velocity was stronger over the front three pressure taps in the upward gust than in the downward gusts, where the change in velocity was most noticeably reduced at the third pressure tap location.

---

alternative shape changes to achieve avian-like aerobatics. Banking, diving, and perching in obstacle-dense environments, such as forests, opens the door to mission performance in natural disaster scenarios such as flooding, hurricanes, and wildfires[54,55]. The extended range offered by adaptive FBF morphing UAVs will greatly improve survey coverage and search and rescue response by increasing the distance covered and time in flight between charges.

## Methods
### Morphing wing construction
We designed the morphing wing with three 42 mm wide active sections separated by two 51 mm wide passive sections to form a 228 mm wide wing with a 320 mm chord. To construct the active sections, we followed the methods established in previous work, which combine a NACA0012 leading edge with an antagonistic double macro-fiber composite (MFC) unimorph trailing edge[17]. We used multi-material 3D printing to include a flexure box design at the interface between the rigid and morphing portion of our active wing section to maximize deflection potential. Unlike in the previous work, we used narrower M8528-P1 MFCs to allow for three active sections to fit within our wind tunnel. Using epoxy, we bonded each MFC to a 0.025 mm stainless steel shim to produce a bending shape change when actuated. We also used epoxy to attach the active trailing edge section to the flexure box interface at the rear of the rigid leading edge.

We constructed the passive sections following methods established by Pankonien et al. for a spanwise morphing wing[17]. The passive sections contain a rigid NACA0012 leading section, but don't have a rigidly structured trailing end. Instead, structure was provided by the spanwise skin extending across the full wing. Bonding a soft 3D-printed mixed cruciform honeycomb to the elastic silicon skin provided additional strength to the

trailing edge of the passive sections[56,57]. This allowed the passive sections to smoothly morph with the active sections while maintaining structural integrity under out of plane aerodynamic loading.

Within each passive section of the wing, we installed six 0.5 mm pressure taps for state observation. The pressure taps were located at positions of 0%, 1.5%, 5%, 10%, 40%, and 50% of the chord length measured from the leading edge. We offset the front four pressure taps at an angle of 30° from the leading tap to mitigate the effect of upstream pressure taps on the flow[58]. Due to the large separation between the front four and rear two pressure taps, we installed the two rearmost pressure taps at a separate 30° angle, not including the front four taps to allow all taps to fit within the passive wing section. Each 1.5 mm pressure tap hole was included in the 3D-printed NACA0012 leading section of the airfoil. We used epoxy to fasten ethyl vinyl acetate tubing into the pressure tap locations. After installation, we used a razorblade to cut the end of each pressure tap to be flush with the surface of the morphing wing to avoid disrupting the flow over the wing.

### Experiment setup
The final morphing wing design was installed 30 cm behind a gust generator (measured at quarter chord positions) in the 30 cm × 30 cm wind tunnel at the University of Michigan (Fig. 5). We created a gusting environment for three flight configurations (high-lift, medium-lift, low-lift) by using various combinations of morphing wing angles of attack ($\alpha = 10 \pm 1°$, $4 \pm 1°$, $4 \pm 1°$) and flow speeds ($U = 10 \, \text{m s}^{-1}$, $15 \, \text{m s}^{-1}$, $10 \, \text{m s}^{-1}$) as measured ahead of the gust generator (Supplementary Table 1). We included elliptical endplates on the wing to prevent wing tip vortices from forming, limiting this analysis to 2D airfoil effects. We measured the morphing wing's lift using a six-axis ATI Delta load cell mounted at the quarter-chord. Six compact differential low-pressure transducers measured the pressures experienced by the six pressure

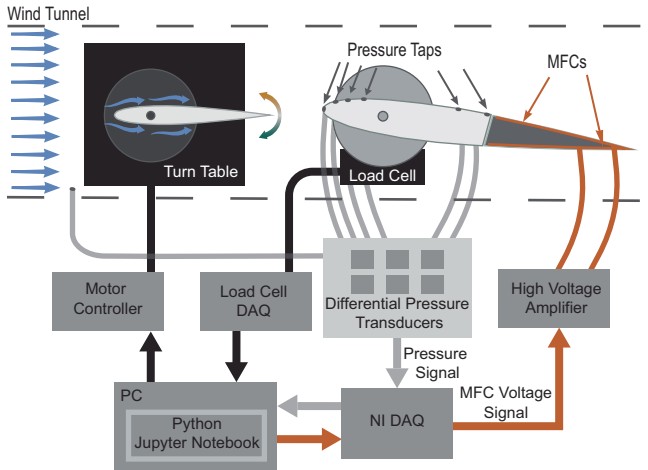

**Fig. 5 | Data flow structure of our gusting wind tunnel experiment for controller training and testing.** Training and testing were orchestrated using a Jupyter Notebook written in Python on a personal computer (PC). The Python script informed the motor controller to rotate the turn table to deflect the gust generator to a desired magnitude and direction. The change in airflow in the wake of the gust generator was detected by the six pressure taps on the macro-fiber composite (MFC) morphing wing. The pressures were measured and compared to a static pressure measured in front of the experimental setup using six differential pressure transducers. Signals from these pressure transducers were acquired by a National Instruments Data Acquisition System (NI-DAQ) and provided to the Python script. The Python script used this information for action selection. The selected action was provided to the NI-DAQ and transformed into an MFC voltage signal which was then amplified to power the MFC camber morphing trailing edge of the wing. The lift produced by the change in camber was measured by the load cell and provided to the Python script for reward calculation during controller training and performance measurement during controller testing.

taps in comparison to the static pressure located at the front of the test section of the wind tunnel, as measured using a pitot-tube. The gust generator consisted of a 15 cm chord NACA0012 rigid wing with a 25 cm span. We used a stepper motor operated turntable to vary the gust generator's angle of attack and create the desired gust deflection.

The gust generator's deflection angle produced different gust intensities depending on the wind tunnel flight condition (high-lift, medium-lift, low-lift). We found the effect of the gust generator setup was sensitive to the angle of attack of our morphing wing. At the highest tested angle of attack $(10 \pm 1°)$, the gust generator produced the smallest effect, even when using larger deflections. We limited our gust generator deflections to a range between positive and negative 12.5° during tests to prevent stall and avoid highly variable wake effects. Training included maximum deflections up to 13.5° to allow for the randomized training exploration to include states around the maximum testing conditions. The generated gusts had greater effect with flight configurations at the lower angle of attack $(4 \pm 1°)$ and gained an even stronger effect at the higher flow speed $(15 \text{ m s}^{-1})$. Therefore, we used gust generator deflection ranges that produced changes in lift that were recoverable within the structural morphing capabilities of the wing (Supplementary Table 1).

To create learned controllers capable of reacting to the changing environment, we adapted an open-source implementation of proximal policy optimization (PPO) in Pytorch to develop policies for the camber morphing wing[59]. The deep reinforcement learning (DRL) environment included a discrete action space. The first testing configuration (high-lift) used a symmetric action space of 7 voltage signal changes. For the subsequent flight conditions (medium-lift and low-lift), we reduced the action space to 3 voltage signal changes, sacrificing potential controller speed for a smaller action space. This compromise required less exploration and potentially improved variability between trained controllers (Supplementary Table 1). Each flight configuration used the same continuous state

space, including normalized change in pressure signals and normalized MFC voltage signals.

The actor and critic network structures included a one-dimensional convolutional neural network input layer with the ten most recent state measurements for state observation, resulting in input dimensions of $2 \times 10$, $4 \times 10$, and $7 \times 10$ for the one-tap, three-tap, and six-tap configurations, respectively (Fig. 6). This layer included convolutions with kernel lengths of three and a stride length of one. The two subsequent hidden layers were structured linearly with 512 nodes each and rectified linear unit (ReLU) activation functions[60,61]. Due to challenges and time constraints associated with DRL training in hardware environments, many hyperparameters were selected based on previous work performed in a similar MFC morphing environment[42] (Supplementary Table 2). However, we tuned the learning rate manually, determining a value of $3 \times 10^{-5}$ to be suitable for Adam optimization[62]. We used change in lift as our optimization parameter, using real-time load cell measurements to provide a reward to the learning algorithm. The goal of the learning algorithm was to develop a controller that minimized the change in lift experienced during a gust using the reward function,

$$R(t) = -10 \times \Delta L_C^2(t). \quad (2)$$

Although lift measurements were used for the reward structure during training, the controllers did not use lift information for action selection. The learned policies only used pressure and MFC voltage signals for action selection. During testing, the load cell provided information to judge controller performance.

A Python script in Jupyter Notebooks orchestrated controller training and testing (Fig. 5). For this work, we defined a gust as a change in effective wind velocity, including speed and direction. Due to electromagnetic interference, the load-cell and pressure sensors were unable to provide accurate signals during step-motor operation. During training and testing, our script paused timestep progression, policy updates, and data collection during gust generator rotation, then resumed training and testing after the gust generator achieved the desired deflection. Due to this full computational pause during rotation, gusts appeared as immediate changes in lift (Fig. 2a). This resulted in perturbations, as viewed by the controller, that are analogous to the sharp-edged updrafts and downdrafts that are used to model changes in lift experienced by small UAVs in gusty city environments[21,23,50,51,63,64].

Training was formatted in a pseudo-episodic manner, alternating between baseline episodes and gusting episodes to facilitate autonomy during training[49]. Each episode began after rotating the gust generator to a specified location depending on the episode's function. Baseline episodes began at zero degrees and gusting episodes began with the gust generator rotated to a random deflection within the specified training gust range (Supplementary Table 1). The MFC actuators began baseline episodes without camber morphing in either direction. From this neutral position, the pressure taps provided a base signal for comparative pressure observations throughout the episode. After initialization was completed, the episode began, including policy action selection and learning updates. The initialized pressure and goal lift values were recorded and carried into the following gusting episode to maintain the same base signals for calculating comparative pressure and reward values. In addition, the MFC sections began gusting episodes actuated to the same position in which they ended the prior baseline episode. Gusting episode action selection and training began after the gust generator was deflected to a randomized position where it was held for the length of the episode, 200 timesteps, representing an extended 10 s gust. Terminating the gust operation represented the completion of a training episode pair (baseline and gusting), returning the gust generator to zero degrees and the morphing wing MFCs to a neutral deflection position to begin a new initialization and subsequent baseline episode.

Training included 1000 total episodes consisting of 200 timesteps of 0.05 s. Learning updates occurred after every 20 timesteps from four

**Fig. 6 | The neural network structure for the actor and critic models in the proximal policy optimization (PPO) algorithm.** Each network has the same base structure, including a 1D convolutional neural network layer followed by three fully connected layers with rectified linear unit (ReLU) actuation functions. The input for each network includes the ten most recent voltage signals supplied to the macro-fiber composites and the ten most recent pressure tap signals, amounting to 2, 4, or 7 measurements for each timestep depending on the pressure tap configuration. The actor network outputs the policy for action selection, including seven actions for the high-lift flight condition and three actions for the medium-lift and low-lift flight conditions. The critic network outputs the state value used to estimate long term expected reward.

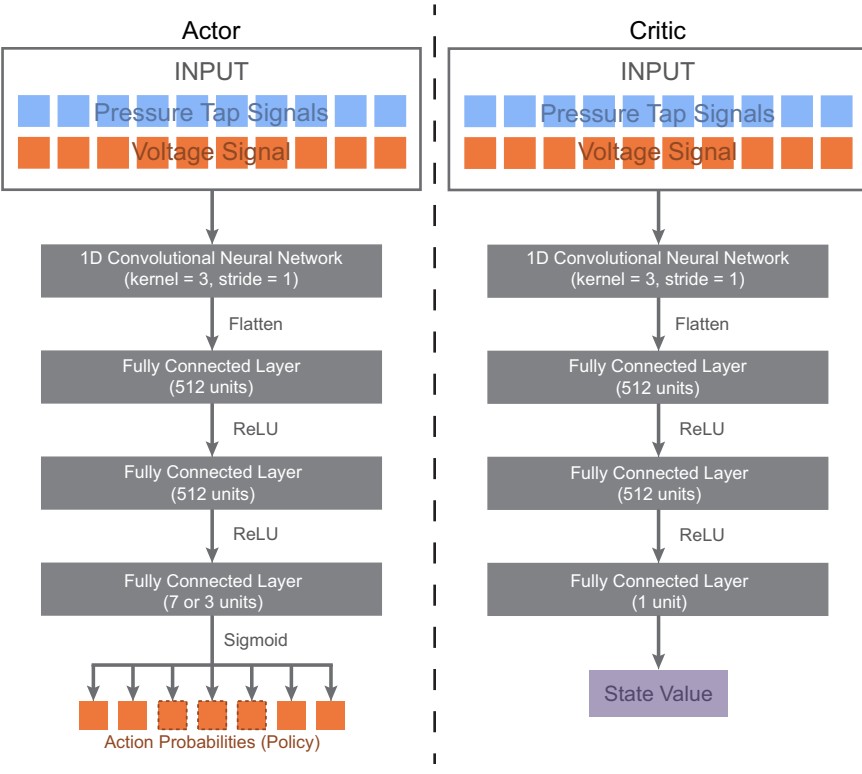

minibatches of five state-action samples in series, resulting in a maximum sample size of $2 \times 10^5$ for policy training. Progress was observed using a running average reward earned over 100 consecutive episodes, from which the highest performing policy was evaluated for testing (Supplementary Fig. 1). We used this procedure to train controllers (high-lift: $n = 10$; medium-lift: $n = 5$; low-lift: $n = 5$) for each of three different pressure tap configurations, including: using all six pressure taps, the front three pressure taps, and a single pressure tap on the leading edge of the morphing wing. We selected these pressure tap configurations based on the pressure distribution expected for the top surface of a symmetric airfoil and the sensitivity of the respective tap locations[58]. In all, this approach resulted in 60 trained controllers.

## Testing
We tested each of the 60 controllers at their trained flight condition (high-lift, medium-lift, low-lift) for three gust magnitudes (mild, moderate, strong) in two directions (upwards and downwards). Upward gusts were denoted as positive and downward as negative (Supplementary Table 1). This resulted in 360 independent testing conditions. Like the baseline training episodes, each testing episode began with an initialization period to reset the base pressure tap signals during neutral airflow. After initialization, the test episode timestep count and controller action selection began. The first quarter of the testing episode consisted of neutral airflow, followed by the gust generator deflecting to a specified gust condition for the following two-quarters of the testing episode. Finally, the gust generator returned to a deflection of zero, concluding the discrete gust and remaining at neutral for the final quarter of the test (Fig. 2a). For each test, we measured controller performance as a gust rejection percentage (GRP), comparing the change in lift experienced by the active camber morphing wing, $\Delta L_C$, to the baseline change in lift measured when the same wing remained unactuated during the gust, $\Delta L_B$ (Eqn. 1) (Fig. 2a).

Due to the black-box nature of neural networks, and the policies developed using such methods, we accounted for stability and robustness of control through repetition. For the initial flight condition (high-lift), we repeated gust alleviation performance tests ten (10) times for each combination of trained controller (10), gust condition (6), and pressure tap configuration (3). This amounted to 1800 gust rejection tests. We measured consistency in performance between test iterations (Supplementary Fig. 5),

gust conditions (Supplementary Fig. 6), and training iterations (Supplementary Fig. 7) while all other factors were held constant. Following the completion of testing at the high-lift flight condition, we repeated the process for five (5) trained controllers at both additional flight configurations (low-lift and medium-lift) to test the robustness of our methods and results for different angles of attack and airflow speeds (Supplementary Table 1). This doubled our previous count of test data, resulting in 3600 gust rejection tests in total (Supplementary Figs. 8–13).

We calculated settled GRP for each gust response test by averaging the GRP achieved during the last half of the gust alleviation test,

$$\text{settled GRP} = \frac{2}{T} \sum_{t=T/2}^{T} \text{GRP}(t). \tag{3}$$

Therefore, a higher settled GRP represented greater gust rejection performance. We calculated the settled GRP values for each individual test, providing distributions of $n = 100$ GRP values for each gust and pressure tap configuration at the high-lift flight condition, and $n = 50$ for each gust and pressure tap configuration at the medium-lift and low-lift flight conditions. Due to the maximum bounded nature of this metric, many distributions were skewed to varying degrees (Supplementary Figs. 14–16). Although the median is traditionally used to represent central tendency for highly skewed distributions, since the distributions were predominantly skewed away from superior performance and there was a large variation in skew between testing conditions, we used the mean as a conservative estimate of central tendency for our primary performance metrics. Further, we use statistical methods to comment on the significance when comparing performances between controllers using different pressure tap configurations. Initially we used a linear mixed effects model to determine the relationship between GRP and the number of pressure taps while considering the random effects of the tested gust conditions and the individual trained controllers. However, we found that the residuals were not normally distributed and therefore broke linear assumptions. Therefore, we trained generalized linear mixed effects models using Markov chain Monte Carlo to provide statistical analyses that were more robust to the variably skewed distributions offered by our tests.

We also considered performance consistency by measuring the absolute difference between the settled GRP of an individual test to the average settled GRP for the associated test condition (flight configuration, gust condition, and number of used pressure taps). This provided a metric for each individual test from which we used another generalized linear mixed effects model to determine significance when comparing gust rejection consistency between controllers using one, three, and six pressure taps.

Finally, we measured the speed of our controllers using rise time, measured as the time required for the learned controllers to increase GRP from 10% to 90% of the settled GRP. Therefore, a lower rise time represented a faster response. Rise times were measured for each test. Although many of these test distributions were highly skewed, because the distributions were predominantly skewed toward slower rise times and there was a large variance in skew between distributions, we again used the mean as a conservative estimate of central tendency (Supplementary Figs. 17–19). Again, we used a generalized linear mixed effects model to analyze the significance between the speed of controllers using one, three, and six pressure taps.

When investigating the sensor signal degradation that occurred during the downward gusts, we used a LaVision particle image velocimetry (PIV) system with DaVis 10 intelligent imaging software to characterize the various aerodynamic effects developed by the gust generator (Fig. 1e). Oil-based smoke particles were accelerated through the open-loop wind tunnel. An EverGreen double-pulse quantel laser mounted outside the wind tunnel illuminated a two-dimensional sheet of particles in the longitudinal dimensions. Above the wind tunnel, two Imager sCMOS cameras in a stereo configuration captured 50 sets of paired images with 15-μs intervals. From this, we captured the mean velocity profiles in the x and z directions of the wind frame of reference up stream of and around the morphing wing, including the locations where pressure taps were installed (Fig. 4c).

## Data availability
All data gathered from experimentation and used for analysis are available to be viewed on the corresponding author's GitHub repository at: https://github.com/kevpatha/few_sensor_gust_alleviation/.

## Code availability
All code used for experimentation and analysis are available to be viewed on the corresponding author's GitHub repository at: https://github.com/kevpatha/few_sensor_gust_alleviation/.

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

## Acknowledgements

This work was supported in part by the National Science Foundation under grant 1935216, as well as the US Air Force Office of Scientific Research under grants numbered FA9550-16-1-0087 and FA9550-21-1-0325.

## Author contributions

K.P.T.H., C.H. and D.J.I. conceived original research idea. K.P.T.H. and C.H. designed the research methodology. K.P.T.H. performed design, fabrication, experimental testing, and data analysis. K.P.T.H. and C.H. organized paper structure and data visualization. D.J.I. performed funding acquisition, project administration, and supervision. K.P.T.H. wrote the original manuscript draft. K.P.T.H., C.H. and D.J.I. revised and edited manuscript.

## Competing interests

The authors declare no competing interests.
