## [Peer Review File · Communications Engineering]

Reviewers' comments:

Reviewer #1 (Remarks to the Author):

The authors develop AI-based "fly-by-feel" control of morphing wing camber for gust rejection. Reinforcement learning trained on wind tunnel gust generation actuates wing camber to optimize gust rejection (GRP) with 1-6 pressure tap inputs. The integration of reinforcement learning, morphing wing design and training/test episodes in physical environments with limited sensing is innovative. With this framework, the authors demonstrate up to 84% gust rejection as formulated by their GRP metric, allowing for systematic variability in trained models, testing and training conditions. Statistical validation is performed by averaging metrics across many test episodes.

However significant questions regarding the methodology should be addressed:

- What is a baseline comparison of this approach, and how do we know 84% GRP is good relative to, for example, corrective gust rejection strategies or model-based control paradigms? Or other fly-by-feel paradigms that required extensive sensor networks?
- More detail on DRL training is needed, including showing the progress of objective over training episode (i.e., a plot of loss vs. iterations), amount of data (samples) to train each policy, input dimension, and any hyperparameter tuning. I am curious why the DRL model consists of two fully connected 512 node layers - this seems to be an excessively overparametrized model that would require significant computational resources and data to train.
- The "Big Data" comment on p4 should be clarified. Training deep learning models is a data-hungry endeavor, not necessarily in the number of inputs (sensors) but rather in the number of samples. Training overparametrized deep learning models requires large numbers of data samples. Following the previous comment, the amount of data to train each policy should be explicitly stated.
- Can the authors elaborate further on the gust generator model? It is stated that the policy training pauses during changes to gust environment. Is the 'discrete square gust column environment' model sufficiently complex to mirror real-world conditions (and hence the trained policy as a result?)

Minor comments:

- What is the sample size for the average GRP in Fig 2?
- p.8 The authors should clarify the duration of each selected policy deployed in the training experimental setup. If "pressure and goal lift values were held for the following episode" where each episode consists of 200 timesteps, how long does it take for policies to activate?
- Given the low latency afforded by fly-by-feel paradigms, mean rise time (from Fig. 3 in the range 0.25-1.25 s) should be emphasized earlier in the text, and compared against "antagonistic" response strategy to gust perturbations (p. 2).

- A major enabler of this work is DRL trained in physical environments. Are there similar successful approaches (DRL trained on physical environments rather than simulation) in other domains (self driving, robotics, biomechanics)?

Reviewer #2 (Remarks to the Author):

This paper considers active control of the lift on a 2D airfoil section using deep reinforcement learning. The control scheme uses a set of pressure sensors as feedback and a deformable aft wing section as the physical control mechanism. Through wing tunnel experimentation, the authors show that gust velocity perturbations can be substantially reduced using the active control system. Specific comments on the paper are provided below.

1. Experimental results are shown for very low lift perturbation levels ($\sim 0.1N$) leading to equivalent angle of attack perturbations of ~ 0.15 deg. These levels are very low and not indicative of what would be required for a practical system. It appears that perturbation levels that are an order of magnitude larger would be in the realm of practical. More practical perturbation levels would make the results much more convincing.
2. One of the stated contributions of the paper is the use of deep learning in combination with pressure feedback. However, no comparisons with existing conventional methods is made to underscore the importance of the authors' approach. It could be that a simple PID controller using an accelerometer would achieve similar results at reduced complexity and cost. The authors should consider comparing their method with existing technology.
3. The gust perturbations that excite the 2D, fixed wind tunnel model are a bit strange. The results only show the streamwise velocity perturbation and larger above and below the airfoil section. The vertical component of the gust is typically much more important and it is important that the gust actually impact the wing section directly. This is probably why the lift perturbations are so small.
4. The time response of the lift perturbation response seems slow (on the order of 0.5 sec). This would more than likely be problematic in an flying aircraft at this scale. In addition, it would be helpful to the reader to see a typical time history of the actuator response (camber time history).
5. Figure 3 should not have a picture of an aircraft. The results in the paper are for a 2D wing section that is fixed in the tunnel. Results are not shown for a flying aircraft.

Dear Reviewers,

Thank you for taking the time to review our manuscript. We have carefully considered all your concerns and have adjusted the manuscript accordingly. We believe the manuscript is now stronger because of your suggestions.

Our responses to the individual concerns are provided below.

Sincerely,

Kevin Haughn

Reviewers' comments:

Reviewer #1 (Remarks to the Author):

The authors develop AI-based "fly-by-feel" control of morphing wing camber for gust rejection. Reinforcement learning trained on wind tunnel gust generation actuates wing camber to optimize gust rejection (GRP) with 1-6 pressure tap inputs. The integration of reinforcement learning, morphing wing design and training/test episodes in physical environments with limited sensing is innovative. With this framework, the authors demonstrate up to 84% gust rejection as formulated by their GRP metric, allowing for systematic variability in trained models, testing and training conditions. Statistical validation is performed by averaging metrics across many test episodes.

However significant questions regarding the methodology should be addressed:

- What is a baseline comparison of this approach, and how do we know 84% GRP is good relative to, for example, corrective gust rejection strategies or model-based control paradigms? Or other fly-by-feel paradigms that required extensive sensor networks?

Thank you for mentioning this. We agree that including a comparative baseline is crucial for showing the impact of our work. We have now added more detailed descriptions of some of the most relevant work we referenced in the paper, including a method using traditional control theory principles, a newer method assuming a preview from a LIDAR system is available, as well as a relevant distributed sensing approach using camber morphing to reject gusts. Additionally, we address the challenges associated with using the more complex methods.

Page 2 Line 15:

However, these corrections occur after the external force has already perturbed the aircraft, and pilot reaction times typically fall between 0.4 and 1.3 seconds after a perturbation signal before providing an input reaction²⁴. This may compromise mission success when strict altitude caps are

in place, such as during nap-of-the-earth flight²⁵. Autopilot systems following classical control theory used traditional control surfaces with strain gauges for feedback to achieve 50% gust load and flight ride quality improvement²³. Recently this response has been improved to 80% when assuming a Doppler light detection and ranging (LIDAR) system was available to provide a preview of incoming gusts²⁶. Instead of responding to a perturbation after it occurs, or spending computation and weight resources on LIDAR systems to look ahead for future perturbations, our fly-by-feel (FBF) active GA senses environmental changes on the wing in real time, beginning the initial morphing reaction in as little as one discrete timestep (0.05) seconds to mitigate unintended changes in aerodynamic forces during a gust.

Successful adaptation, such as that provided by GA, relies on an accurate representation of the changing environment²⁷⁻²⁹. FBF is a biologically inspired paradigm that uses distributed sensors to inform UAVs of environmental changes²⁹⁻³⁴. Recently, FBF achieved up to 76% mean gust rejection on a servo-driven camber morphing wing by using incremental nonlinear dynamic inversion with quadratic programming and virtual shape functions (INDI-QP-V) used sixteen on-board piezoelectric pressure sensors to detect changes in the airflow for state inference, as well as fourteen fiberoptic cables, twelve strain gauges and a wing root mounted camera to detect camber deflection with proprioceptive modeling³⁵. However, the expansive sensing networks used to inform decision making through proprioception and state inference add weight and challenge the computational power capabilities offered by small UAVs^{5-7,28,36}.

- More detail on DRL training is needed, including showing the progress of objective over training episode (i.e., a plot of loss vs. iterations), amount of data (samples) to train each policy, input dimension, and any hyperparameter tuning. I am curious why the DRL model consists of two fully connected 512 node layers - this seems to be an excessively overparametrized model that would require significant computational resources and data to train.

Thank you for pointing out this missing information. We added a significant portion to the experiment setup section of the manuscript including: input dimensions, network size, hyperparameter selection and tuning, learning update frequency and size, as well as the total sample size used in policy training. Additionally, we've included a reference to our first figure in the supplemental materials providing the necessary plots showing growth in average reward earned vs. training episode. Although we use deep reinforcement learning in this work, it was not the primary focus, but rather a tool to investigate design and the use of distributed sensors for fly-by-feel control. The following has been added to the experiment setup section on page 8, including a reference to a new table in the supplemental material including all the hyperparameter values.

Page 8 Line 16

The actor and critic network structures included a one-dimensional convolutional neural network input layer with the ten most recent state measurements for state observation, resulting in input dimensions of 2×10, 4×10, and 7×10 for the one-tap, three-tap, and six-tap configurations respectively (Fig. 6). This layer included convolutions with kernel lengths of three and a stride length of one. The two subsequent hidden layers were structured linearly with 512 nodes each

and rectified linear unit (ReLU) activation functions^{61,62}. Due to challenges and time constraints associated with DRL training in hardware environments, many hyperparameters were selected based on previous work performed in a similar MFC morphing environment⁵² (Supplementary Table 2). However, we tuned the learning rate manually, determining a value of 3×10^{-5} to be suitable for Adam optimization⁶³.

And Page 9 Line 19

Training included 1000 total episodes consisting of 200 timesteps of 0.05 seconds. Learning updates occurred after every 20 timesteps from four minibatches of five state-action samples in series, resulting in a maximum sample size of 2×10^5 for policy training. Progress was observed using a running average reward earned over 100 consecutive episodes, from which the highest performing policy was evaluated for testing (Supplementary Fig. 1).

- The "Big Data" comment on p4 should be clarified. Training deep learning models is a data-hungry endeavor, not necessarily in the number of inputs (sensors) but rather in the number of samples. Training overparametrized deep learning models requires large numbers of data samples. Following the previous comment, the amount of data to train each policy should be explicitly stated.

Thank you for bringing this lack of clarity to our attention. We agree that this is an important concept to highlight. We have updated the text to better highlight the newly growing tendency in the aerospace engineering field to use deep learning networks and their usefulness when using large amounts of input data to overcompensate for a perceived lack of information with an oversized sensing network to achieve fly-by-feel. For this reason, we removed that portion of the results section and added the following to the discussion section to improve clarity.

Page 6 Line 7

These results run counter to the big data mentality that is pervasive in deep learning and has recently driven sensor network design in machine learning based distributed sensing applications, including FBF^{30,31,53}. Like intelligent feature selection in deep learning, intelligent controller and sensor design can achieve reduced-sensor FBF, providing an efficient alternative to large scale distributed sensing networks⁵⁴. This reduces mechanical complexity and cost during fabrication as well as weight and computational requirements during operation.

- Can the authors elaborate further on the gust generator model? It is stated that the policy training pauses during changes to gust environment. Is the 'discrete square gust column environment' model sufficiently complex to mirror real-world conditions (and hence the trained policy as a result?)

Thank you for bringing this to our attention. We have significantly adjusted the description to our experiment setup section.

Page 8 Line 41

During training and testing, our script paused timestep progression, policy updates, and data collection during gust generator rotation, then resumed training and testing after the gust generator achieved the desired deflection. Due to this full computational pause during rotation, gusts appeared as immediate changes in lift (Fig. 2a). This resulted in perturbations, as viewed by the controller, that are analogous to the sharp-edged updrafts and downdrafts that are used to model changes in lift experienced by small UAVs in gusty city environments^{21,23,50,51,64,65}.

Training was formatted in a pseudo-episodic manner, alternating between baseline episodes and gusting episodes to facilitate autonomy during training⁴⁹. Each episode began after rotating the gust generator to a specified location depending on the episode's function. Baseline episodes began at zero degrees and gusting episodes began with the gust generator rotated to a random deflection within the specified training gust range (Supplementary Table 1). The MFC actuators began baseline episodes without camber morphing in either direction. From this neutral position, the pressure taps provided a base signal for comparative pressure observations throughout the episode. After initialization was completed, the episode began, including policy action selection and learning updates. The initialized pressure and goal lift values were recorded and carried into the following gusting episode to maintain the same base signals for calculating comparative pressure and reward values. Additionally, the MFC sections began gusting episodes actuated to the same position in which they ended the prior baseline episode. Gusting episode action selection and training began after the gust generator was deflected to a randomized position where it was held for the length of the episode, 200 timesteps, representing an extended 10 second gust. Terminating the gust operation represented the completion of a training episode pair (baseline and gusting), returning the gust generator to zero degrees and the morphing wing MFCs to a neutral deflection position to begin a new initialization and subsequent baseline episode.

And we added the following to the Testing section.

Page 9 Line 35

Like the baseline training episodes, each testing episode began with an initialization period to reset the base pressure tap signals during neutral airflow. After initialization, the test episode timestep count and controller action selection began.

Minor comments:

- What is the sample size for the average GRP in Fig 2?

Thank you for pointing this out, we have added the sample count to the caption on Fig 2.

Page 17 Line 8

b) On average ($n = 600$), the learned controllers rejected more than 84% of the ΔL_B produced by the tested gusts.

- p.8 The authors should clarify the duration of each selected policy deployed in the training experimental setup. If "pressure and goal lift values were held for the following episode" where each episode consists of 200 timesteps, how long does it take for policies to activate?

We have made the necessary adjustments to address this concern in the previous response regarding experimental setup.

- Given the low latency afforded by fly-by-feel paradigms, mean rise time (from Fig. 3 in the range 0.25-1.25 s) should be emphasized earlier in the text, and compared against "antagonistic" response strategy to gust perturbations (p. 2).

This is a good point, we have added the following statement to address this in the introduction.

Page 2 Line 15

However, these corrections occur after the external force has already perturbed the aircraft, and pilot reaction times typically fall between 0.4 and 1.3 seconds after a perturbation signal before providing an input reaction²⁴.

Page 2 Line 24

our fly-by-feel (FBF) active GA senses environmental changes on the wing in real time, beginning the initial morphing reaction in as little as one discrete timestep (0.05) seconds to mitigate unintended changes in aerodynamic forces during a gust.

Additionally, we have provided more depth to the results regarding time response and moved it to be earlier in the results section, so now it directly follows paragraphs discussing gust rejection.

Page 4 Line 42

Timing is also a crucial component of perturbation response since a slower reaction would negate much of the benefit offered by the correction. The instantaneous change in lift produced by the sharp-edged gusting environment neglected the buildup in gust intensity typically found in nature, creating a challenging environment for controller response. Still, using rise time, we quantified the controllers' speed to comment on how reducing sensor count impacted the active responsiveness of the system (Fig. 3e-h). We found that the controller speed was not significantly affected by the pressure tap configurations ($P > 0.05$) for all flight conditions and was consistent with rise times established in previous work where DRL controllers showed to be faster than traditional feedback control methods for an MFC morphing wing⁵² (Fig. 3f-h). However, the higher intensity gusts resulted in greater rise times, which suggests the limited discrete action space likely restricted controller speeds. Rise time uncertainty was considered using standard deviation, as done previously with GRP (Supplementary Fig. 3d-f).

Finally, we have added a brief portion to the discussion sentence to define our findings more explicitly with respect to controller speed.

Page 6 Line 13

Additionally, where human pilots naturally have a delayed initial reaction (0.4 to 1.3 seconds) to gusts, FBF can begin changing shape in as little as a single timestep (0.05 seconds), and we showed that the controller speed was not impacted by reduced sensor input²⁴. Further, we expect that optimizing the controller action space would provide a more rapid response.

- A major enabler of this work is DRL trained in physical environments. Are there similar successful approaches (DRL trained on physical environments rather than simulation) in other domains (self driving, robotics, biomechanics)?

This is an excellent point. Performing DRL in a physical environment does produce many challenges, that have been addressed in the literature (now cited). Additionally, we have specified our use of a pseudo-episodic based learning method to address these challenges, adding clarity to the manuscript.

Page 3 Line 4

Most successful DRL applications are trained in simulation due to the repetitive nature of DRL's trial-and-error training format^{44,45}. However, accurately simulating complex, gusty environments requires large computational time and cost^{46,47}. We avoided the computational costs as well as the uncertainty associated with simplified approximation by training directly on the physical hardware environment. Although training in the physical hardware space offers unique challenges, we found success using methods emphasizing efficiency and autonomy in state-action exploration through a pseudo-episodic training method^{48,49}.

Reviewer #2 (Remarks to the Author):

This paper considers active control of the lift on a 2D airfoil section using deep reinforcement learning. The control scheme uses a set of pressure sensors as feedback and a deformable aft wing section as the physical control mechanism. Through wing tunnel experimentation, the authors show that gust velocity perturbations can be substantially reduced using the active control system. Specific comments on the paper are provided below.

1. Experimental results are shown for very low lift perturbation levels ($\sim 0.1N$) leading to equivalent angle of attack perturbations of ~ 0.15 deg. These levels are very low and not indicative of what would be required for a practical system. It appears that perturbation levels that are an order of magnitude larger would be in the realm of practical. More practical perturbation levels would make the results much more convincing.

Thank you for bringing to our attention that we did not clearly present the various flight conditions in which our controllers were tested. It is crucial that our readers understand that we trained and tested gust rejection controllers in realistic gust conditions, and for this reason we tested three different environments (denoted as High-lift, Medium-lift, and Low-lift). It is true that the first flight condition

(High-lift) produced smaller magnitude changes in lift (2%-5%). However, this was due to the wing's set angle of attack for that condition. The change in lift produced at the other flight conditions (Med-lift and Low-lift) were significantly larger (10%-28% and 11%-29%) creating realistic, and even large, gust conditions for the more traditional flight configurations (table S1). To clarify this, we now more clearly draw attention to this difference between the tested gust condition and have included the gust magnitudes in the paper directly.

Page 3 Line 38

To replicate common scenarios experienced during city flight, tests were conducted at three different flight conditions (low-lift, medium-lift, and high-lift) for three gust magnitudes (mild, moderate, and strong) in two directions (upward and downward) (Supplementary Table 1). Although the high-lift condition experienced smaller gust impact (5% change in lift), the medium-lift and low-lift conditions experienced much larger ranges and magnitudes of gust impacts (28% and 29% change in lift respectively).

Additionally, we've referenced a new PIV figure in the supplementary material at the beginning of the results section to show the local change in angle of attack reached perturbations of over 10 degrees.

Page 3 Line 25

The gust generator used in this wind tunnel environment perturbed the local angle of attack for the incoming airflow in a manner analogous to common flight situations in natural and urban environments (Supplementary Fig. 2).

2. One of the stated contributions of the paper is the use of deep learning in combination with pressure feedback. However, no comparisons with existing conventional methods is made to underscore the importance of the authors' approach. It could be that a simple PID controller using an accelerometer would achieve similar results at reduced complexity and cost. The authors should consider comparing their method with existing technology.

Thank you for bringing up this point. We agree that this point is crucial for us to better highlight in our manuscript. Other, more traditional control methods, require accurate models for inferring the state of the system or predicting the dynamics. Previous work has shown that achieving accurate proprioception of the MFC camber morphing wing is not a trivial task due to nonlinearities including hysteresis and creep and requires additional internal sensors and nonlinear modeling for feedback [Haughn, 2022]. Therefore, combining the nonlinear proprioceptive modeling with the exteroceptive state inference for this complex aerodynamic environment greatly complicates the problem for implementing a more traditional feedback controller. However, it was shown that DRL can perform well in these systems even with imperfect feedback, and it was with this motivation that we carried forward using only direct sensor signal based decision making with DRL [Haughn, 2023]. Implementing a traditional feedback controller using accelerometer feedback would not work for our experimental setup as the wing is fixed in the wind tunnel and unable to perform rigid body motion, and using the force sensor for direct feedback on the system would not provide a fair comparison since a key part of this work was to avoid using perfect feedback.

However, we agree that providing a baseline comparison is important, and have therefore included a more thorough reference to the state of the art found in literature and emphasized our point by in the introduction.

Page 2 Line 19

Autopilot systems following classical control theory used traditional control surfaces with strain gauges for feedback to achieve 50% gust load and flight ride quality improvement²³. Recently this response has been improved to 80% when assuming a Doppler light detection and ranging (LIDAR) system was available to provide a preview of incoming gusts²⁶. Instead of responding to a perturbation after it occurs, or spending computation and weight resources on LIDAR systems to look ahead for future perturbations, our fly-by-feel (FBF) active GA senses environmental changes on the wing in real time, beginning the initial morphing reaction in as little as one discrete timestep (0.05) seconds to mitigate unintended changes in aerodynamic forces during a gust.

Successful adaptation, such as that provided by GA, relies on an accurate representation of the changing environment²⁷⁻²⁹. FBF is a biologically inspired paradigm that uses distributed sensors to inform UAVs of environmental changes²⁹⁻³⁴. Recently, FBF achieved up to 76% mean gust rejection on a servo-driven camber morphing wing by using incremental nonlinear dynamic inversion with quadratic programming and virtual shape functions (INDI-QP-V) used sixteen on-board piezoelectric pressure sensors to detect changes in the airflow for state inference, as well as fourteen fiberoptic cables, twelve strain gauges and a wing root mounted camera to detect camber deflection with proprioceptive modeling³⁵. However, the expansive sensing networks used to inform decision making through proprioception and state inference add weight and challenge the computational power capabilities offered by small UAVs^{5-7,28,36}.

3. The gust perturbations that excite the 2D, fixed wind tunnel model are a bit strange. The results only show the streamwise velocity perturbation and larger above and below the airfoil section. The vertical component of the gust is typically much more important and it is important that the gust actually impact the wing section directly. This is probably why the lift perturbations are so small.

Thank you for bringing this to our attention. Although we do show the wake of the gust generating airfoil as a streamwise change in velocity (Fig 1e) the PIV in Figure 4C is depicting the overall velocity magnitude specifically to comment on pressure tap sensitivity. We had originally used the streamwise velocity, but switched to the total magnitude since it is a better representation of what the sensors would be experiencing but forgot to update the color bar label. Thank you for catching that mistake, it has been fixed.

Regarding the gust generation setup, we used a method that has been used similarly in other gust generating experimental setups found in the literature and could be autonomously reproduced for DRL training. We have added greater description of our gust generating setup and highlighted the use of an established method in our manuscript.

Page 3 Line 10

This training format requires an automatic transition between episodes. Therefore, we adapted methods previously established in the literature to automate a gusting environment^{29,35,36}. By deflecting a rigid wing, mounted in a wind tunnel upstream of our morphing wing, we exposed the morphing wing to a broad range of repeatable gusts during training to facilitate thorough exploration of the dynamic environment's state and action spaces (Fig. 1d).

Additionally, we have included a new figure in the Supplementary material showing the change in local angle of attack (Supplementary Fig. 2).

4. The time response of the lift perturbation response seems slow (on the order of 0.5 sec). This would more than likely be problematic in an flying aircraft at this scale. In addition, it would be helpful to the reader to see a typical time history of the actuator response (camber time history).

Thank you for mentioning this concern. Since a significant benefit of this reduced sensor DRL control method was to achieve morphing gust rejection without proprioceptive sensing, we did not have sensors nor data available to represent true camber of the system or actuator response. However, previous work by the authors has focused on actuator response (with camber time history) and has now been sufficiently referenced in the manuscript.

Page 4 Line 47

We found that the controller speed was not significantly affected by the pressure tap configurations ($P > 0.05$) for all flight conditions and was consistent with rise times established in previous work where DRL controllers showed to be faster than traditional feedback control methods for an MFC morphing wing⁵² (Fig. 3f-h). However, the higher intensity gusts resulted in greater rise times, which suggests the limited discrete action space may have restricted controller speeds.

Additionally, the step like response of our environment neglects the typical buildup of changing lift found in nature but provides the opportunity to use the rise time to compare controller speed between the varied pressure tap configurations. Therefore, we've added clarification for our use of rise time. measured in this work is relevant due to the step function like behavior.

Page 4 Line 42

Timing is also a crucial component of perturbation response since a slower reaction would negate much of the benefit offered by the correction. The instantaneous change in lift produced by the sharp-edged gusting environment neglected the buildup in gust intensity typically found in nature, creating a challenging environment for controller response. Still, using rise time, we quantified the controllers' speed to comment on how reducing sensor count impacted the active responsiveness of the system (Fig. 3e-h).

We agree that the overall performance expectation should still be discussed and have added more to the discussion section to draw perspective to piloted systems as well.

Page 6 Line 13

Additionally, where human pilots naturally have a delayed initial reaction (0.4 to 1.3 seconds) to gusts, FBF can begin changing shape in as little as a single timestep (0.05 seconds), and we showed that the controller speed was not impacted by reduced sensor input²⁴. Further, we expect that optimizing the controller action space would provide a more rapid response.

5. Figure 3 should not have a picture of an aircraft. The results in the paper are for a 2D wing section that is fixed in the tunnel. Results are not shown for a flying aircraft.

Thank you, we believe that this figure represents the possible application of this work and therefore have left it in. However, we have added labeling to clarify its purpose as an illustrative example of real-world flight conditions we are replicating.

REVIEWERS' COMMENTS:

Reviewer #1 (Remarks to the Author):

The authors have satisfactorily addressed referee comments.